# The Public Health Importance and Management of Infectious Poultry Diseases in Smallholder Systems in Africa

**DOI:** 10.3390/foods13030411

**Published:** 2024-01-26

**Authors:** Delia Grace, Theodore J. D. Knight-Jones, Achenef Melaku, Robyn Alders, Wudu T. Jemberu

**Affiliations:** 1Natural Resources Institute (NRI), Chatham ME4 4TB, UK; 2International Livestock Research Institute (ILRI), Nairobi P.O. Box 30709, Kenya; 3International Livestock Research Institute (ILRI), Addis Ababa P.O. Box 5689, Ethiopia or wudu.temesgen@uog.edu.et (W.T.J.); 4Department of Veterinary Pharmacy, University of Gondar, Gondar P.O. Box 196, Ethiopia; achenefmela@yahoo.com; 5Development Policy Centre, Australian National University, Acton, Canberra 2601, Australia; robyn.alders@anu.edu.au; 6Department of Veterinary Epidemiology and Public Health, University of Gondar, Gondar P.O. Box 196, Ethiopia

**Keywords:** poultry, backyard, Africa, infectious, disease, zoonoses, public health

## Abstract

Poultry diseases pose major constraints on smallholder production in Africa, causing high flock mortality and economic hardship. Infectious diseases, especially viral diseases like Newcastle disease and highly pathogenic avian influenza (HPAI) and bacterial diseases, especially colibacillosis and salmonellosis, are responsible for most chicken losses, with downstream effects on human nutrition and health. Beyond production impacts, poultry diseases directly harm public health if zoonotic, can give rise to epidemics and pandemics, and facilitate antimicrobial resistance through treatment attempts. HPAI, campylobacteriosis, and salmonellosis are the priority zoonoses. Sustainable solutions for poultry health remain elusive despite recognition of the problem. This review summarises current knowledge on major poultry diseases in smallholder systems, their impacts, and options for prevention and control. We find biosecurity, vaccination, good husbandry, and disease-resistant breeds can reduce disease burden, but practical limitations exist in implementing these measures across smallholder systems. Treatment is often inefficient for viral diseases, and treatment for bacterial diseases risks antimicrobial resistance. Ethnoveterinary practices offer accessible alternatives but require more rigorous evaluation. Multisectoral collaboration and policies that reach smallholder poultry keepers are essential to alleviate disease constraints. Successful control will improve livelihoods, nutrition, and gender equity for millions of rural families. This review concludes that sustainable, scalable solutions for smallholder poultry disease control remain a critical unmet need in Africa.

## 1. Introduction

This comprehensive review aims to provide an in-depth understanding of the public health implications of important small-scale poultry diseases in Africa and how these diseases can be managed to reduce human health impact. As the focus is on smallholder systems, including scavenging extensive systems and small-scale semi-intensified or intensified systems, most of the literature is from sub-Saharan Africa where smallholder systems predominate, despite the rapid growth of intensive systems. We consider broad public health implications that are not limited to zoonoses but include antimicrobial resistance and food and nutrition security.

This review starts with the context of smallholder poultry systems and the importance of poultry disease: public health impacts are emphasised, including nutrition. It then highlights common diseases in smallholder poultry, their impact on income, food and nutrition security, and public health. Next, preventive and control measures to reduce the burden of poultry diseases are discussed. The review begins by examining the prevalence of diseases commonly observed in smallholder poultry. It then delves into general health management strategies, followed by an exploration of specific diseases and their impacts. Diseases with public health implications are given particular attention. Additionally, the review covers the role of ethnoveterinary medicine in poultry health. A brief conclusion summarises the key findings. The information presented here aims to assist policymakers, animal health practitioners, livestock officers, and extension agents in improving the health and productivity of smallholder poultry production across Africa.

## 2. Context

Smallholder poultry production plays a crucial role in Africa, especially sub-Saharan Africa, providing numerous important benefits. It serves as a source of income and livelihood for small-scale farmers, particularly women and the youth, contributing to poverty reduction and economic empowerment. It contributes to food and nutrition security, as poultry products are a valuable source of animal protein and essential micronutrients. Additionally, smallholder poultry farming is often more environmentally sustainable compared to larger commercial operations as it utilises local resources, requires less infrastructure, has a lower carbon footprint, and indigenous poultry breeds are more resilient to extreme weather events [1].

Poultry population density is highest in coastal West Africa, Ethiopia, the Great Lakes region, and Southern Africa (Figure 1). Family poultry comprises up to 80% of poultry stocks in sub-Saharan Africa but only around 50% of meat consumption [2]. Many poor households in Africa rarely eat poultry products. In Kigali, the capital of Rwanda, chicken only 0.8% of households prepared chicken dishes on a weekly basis [3]. In rural Tanzania, most households consumed chicken only on special occasions, while eggs were eaten never or occasionally [4]. In Ethiopia, typical low-income urban households ate chicken about once every three months [5].

Yet despite the small amounts consumed, poultry products make vital contributions to food and nutrition security [4]. Like other animal source foods, they provide high-quality protein and micronutrients in bioavailable forms which, even in small quantities, substantially increase the nutrient adequacy of traditional diets based on staple crops. Poultry meat is recognised as a healthy source of protein, and the older, firmer, and more flavoursome meat of indigenous birds commonly kept by smallholders is often preferred by African consumers and commands a price premium. Eggs are important sources of choline, vitamins, and high-biological-value protein. A 50 g edible portion of egg provides nearly half the recommended daily allowances of protein for children of 1–3 years of age, including all essential amino acids [7].

However, smallholder poultry production and the current and potential benefits of poultry products to food and nutrition security face significant challenges in Africa due to various poultry health problems [8]. Among these, diseases are consistently identified as the primary constraint within the sector [9]. This contrasts with high-input systems, where feed costs dominate. At the individual animal level, diseases contribute to reduced productivity, manifested as decreased output per unit of inputs, and can result in morbidity and mortality, leading to a complete loss of production and production assets. In addition to these economic impacts, diseases have broader implications for public health.

Firstly, disease losses both reduce the availability of nutritious meat and eggs and constrain the potential growth of the poultry section. This can have significant impacts in vulnerable communities, as was documented when a highly pathogenic avian influenza outbreak resulted in the mass culling of chickens in Lower Egypt but not Upper Egypt. Decreased dietary diversity, reduced poultry consumption, the substitution of nutritious foods with sugary foods, and increased stunting were seen in Lower Egypt but not Upper Egypt [10].

Secondly, some diseases are zoonotic, meaning they can naturally transmit from animals to humans through food or contact, resulting in human illness and economic losses. Certain zoonotic diseases, such as avian influenza, have the potential to cross over to humans and mutate into forms that are more easily transmitted between people, with potentially significant human health and economic consequences.

Finally, because farmers often respond to disease or concern that diseases are spreading locally by using antimicrobials, even non-zoonotic poultry diseases indirectly contribute to antimicrobial resistance in both people and animals [11,12].

## 3. The Importance of Infectious Disease

Disease is generally considered as a harmful deviation from normal function or structure, associated with specific clinical signs (or in the case of humans, symptoms). Disease may be caused by pathogenic organisms, toxic agents, nutritional deficiencies, metabolic abnormalities, neoplasia, genetic anomalies, or injuries (including predation). Smallholder poultry production in Africa is characterised by high levels of disease, resulting in high mortality that can reach up to 50% in the general population and 75% in brooding chicks (up to the age of 8 weeks) [12,13,14,15,16]. For example, in Ethiopia in 2020, the premature death of 39 million chickens out of a national population of 57 million was reported [17], and a multiyear longitudinal study in Western Kenya showed about 60% of offtake in village chickens is due to mortality from disease [15].

Smallholder poultry production in Africa is particularly susceptible to high levels of disease, for a number of reasons. Firstly, the environment where smallholder poultry are farmed often lacks adequate sanitation and appropriate biosecurity measures, facilitating the transmission and spread of diseases. Additionally, the availability of inputs, such as quality feed and clean water, may be limited or of poor quality, compromising the overall health and immunity of the poultry. Moreover, smallholder farmers may have limited access to veterinary services, diagnostic tools, and preventive measures, leading to inadequate disease prevention, control, and management. Furthermore, many smallholder farmers may have limited knowledge and resources to implement appropriate biosecurity protocols and effective management practices, further increasing the vulnerability of their poultry to diseases.

The widespread presence of diseases not only imposes an economic burden on traditional village production systems but also hinders their transformation into more productive, efficient, and sustainable systems [8]. The introduction of commercial breeds, cross-bred or hybrid chickens, and enhanced management practices is often impeded by disease prevalence [18]. Furthermore, “improved” breeds, primarily selected for their production traits, tend to be more susceptible to diseases compared to local varieties, which possess greater genetic diversity and natural resistance to diseases [19]. Efforts to improve smallholder poultry production have had limited success in the past, largely due to the high mortality experienced by exotic breeds. As a result, many current programs distribute hybrid chickens, but this causes multiple problems when farmers try to breed them. Among the available interventions, vaccination has shown the greatest impact in terms of improving flock productivity [20].

The risk of poultry health problems or actual disease outbreaks can lead to the abandonment of chicken production by smallholder producers. It is worth noting that young people and women are often heavily involved in smallholder poultry production in low- and middle-income countries (LMICs), so they are disproportionately impacted by poultry health problems. Therefore, improving poultry health would not only enhance livelihoods and nutrition but also have a positive impact on marginalised members of society, particularly women and children. However, animal health services, especially for smallholder poultry, are poorly developed in many LMICs, resulting in limited access to diagnostic, preventive, and treatment services for producers [21].

## 4. The Public Health Impacts of Poultry Disease in Smallholder Systems

Non-zoonotic poultry diseases resulting in reduced productivity and decreased income affect public health by reducing food and nutrition security [13,22,23]. More direct public health effects are due to zoonotic diseases which are transmitted from poultry to humans through food, water, and direct contact. Certain viral diseases, such as influenzas, are of great importance because of their pandemic potential. The 1918 “Spanish flu” pandemic, which killed up to 100 million people worldwide, most likely had an avian origin [24]. In Africa, nearly a third of children are stunted (or too short for their age), which indicates chronic under-nutrition and is associated with increased vulnerability to disease, poor cognitive development, and poor life outcomes [25]. *Campylobacter* spp. are among the main pathogenic bacteria implicated in stunting [26]. Children mainly acquire *Campylobacter* through the consumption of contaminated poultry meat or by exposure to faeces from poultry and other livestock species (including the ingestion of faeces on fomites or on the ground) [27]. The emergence of antimicrobial resistance (AMR) is a global health problem thought to be directly responsible for 1.27 million human deaths each year only for bacterial AMR [28]. The use of antimicrobials in animals fosters AMR, which can then transfer to humans. AMR may be an increasing problem with the growing intensification of the poultry sector in low- and middle-income countries [11]. There has been little research on the psycho-social impacts of livestock disease on African farmers, but a recent study in Ghana found infectious diseases were the most common cause of livestock losses and this negatively affected the mental health of farmers [29].

Smallholder free-range or semi-free-range poultry also serve as a pathogen reservoir for intensive poultry [30] and a disease interface between wild birds and more biosecure, intensive systems. The latter transmission pathway is especially important for HPAI. In several countries, smallholder poultry farms are reported to have a higher incidence of HPAI and a higher proportion of disease outbreaks, suggesting they play a role in maintaining infection [31]. Recently, it has become apparent that village chickens may not only serve as a biosecurity risk to commercial flocks but also as a reservoir for the spillover of common “poultry” pathogens to wild birds, although the latter is harder to detect. However, in Nigeria, phylogenetic analysis of Newcastle disease virus showed transmission from domestic poultry to black kite [32].

## 5. The Prevalence and Impacts of Diseases on Smallholder Poultry Production

The term “disease” encompasses a range of conditions, including injuries and poisonings [33]. Diseases can be categorised into infectious diseases, which include viral, bacterial, and parasitic diseases, and non-infectious or non-communicable diseases, such as nutritional, metabolic, genetic, and neoplastic diseases, as well as injuries and poisonings. Infectious diseases often pose the greatest economic and public health burden, although predation can also cause significant losses in free-ranging and scavenging poultry systems. While nutritional issues can lead to poor growth and increased susceptibility to disease, the dietary deficiencies and toxicoses commonly observed in intensively managed chickens are less prevalent in smallholder poultry production, especially in backyard and scavenging birds. However, poisonings, such as snakebite envenomation, may occur more frequently in extensively managed smallholder poultry and are probably underestimated. A study conducted in Cameroon reported that out of 332 recorded cases of snakebite in livestock, poultry accounted for the majority with 207 cases (62%) [34]. In some instances, envenomed birds were subsequently eaten by predators. While non-infectious diseases are of great importance, this review will specifically focus on infectious diseases of poultry.

Transmissible infectious diseases pose a significant threat to smallholder poultry in Africa, with examples including Newcastle disease and avian influenza [35,36]. Both are zoonoses, although Newcastle disease is of minor zoonotic importance, and the main public health risk of avian influenza is not its impact as a directly transmitted zoonosis, but the risk that it will evolve to a strain that is transmissible between people. Both diseases can cause explosive outbreaks with high mortality rates, leading to devastating losses for smallholder farmers. Other infectious diseases commonly reported in smallholder poultry systems include those caused by endoparasites and ectoparasites, causing ongoing losses [37,38]. Zoonotic diseases, which can be transmitted from animals to humans, also pose serious risks to public health.

In addition to Newcastle disease and avian influenza, serological studies frequently find antibodies to viral diseases in smallholder poultry such as infectious bursal disease (IBD) and Marek’s disease. However, the presence of IBD antibodies is indicative of infection only and not necessarily clinical disease. (Most studies reporting IBD are serological.) IBD is unlikely to have a major impact on multi-age indigenous chicken flocks. Bacterial diseases such as fowl typhoid and fowl cholera can also impact smallholder poultry production. Furthermore, parasitic diseases like coccidiosis, helminth infestation, and ectoparasite infestation are prevalent and contribute to production losses. Although there are numerous other infectious diseases, research often focuses on a few high-impact ones.

A systematic literature review study conducted in Ethiopia revealed that 14 infectious and parasitic diseases were reported in 110 studies published from 2000 to 2017. Among these, Newcastle disease, IBD, avian coccidiosis, helminth infestation, ectoparasite infestation, and salmonellosis were the most frequently studied diseases [39]. Eight of the diseases (more than half) were also zoonoses. Many of these diseases have also been identified as major constraints on smallholder chicken production in the broader East Africa region [40]. Table 1 summarises the most common and serious production diseases for smallholder chickens in Africa.

Most smallholder poultry keepers in Africa do not have good access to veterinary services and so cases of disease are usually not confirmed and not reported, with limited progress in terms of improving the surveillance of important diseases to inform control efforts [41].

## 6. Priority Zoonotic Poultry Diseases of Smallholder Poultry of Public Health Significance

Poultry zoonoses of major public health concern fall into two categories. First are emerging diseases which have the potential to undergo mutations that allow them to shift from poultry-to-human transmission to human-to-human transmission, which can result in epidemics or pandemics. Avian influenza is a prototype for these emerging viruses. The second category of major public health importance are zoonoses which cause a high number of human illnesses: the most important of these are foodborne (Table 2).

Avian influenza, caused by a type A influenza virus, is a significant global disease in poultry and a priority human-emerging infectious disease. It affects both domestic and wild birds, and certain strains can also be transmitted to humans, making it a zoonosis. Avian influenza has two forms: highly pathogenic avian influenza (HPAI) with severe disease and high mortality in domestic chickens, and low-pathogenicity avian influenza (LPAI) with milder symptoms in chickens, but LPAI strains can sometimes mutate into HPAI. HPAI outbreaks have occurred in many countries in Africa, particularly in western and southern regions, often linked to migrating waterfowl and then spreading within domestic poultry [35,43]. Wild waterfowl act as reservoirs for avian influenza but do not typically show severe signs of illness. Studies estimate an overall prevalence of avian influenza in sub-Saharan Africa to be around 3% based on viral isolation and genome detection and 4% based on antibody detection [35]. However, these studies may not capture the full extent of the disease’s impact, as high-mortality cases are not always detected. A study in Mali found a low incidence rate of 0.7 birds per 100 bird months at risk and a seroprevalence of 2.9% for avian influenza in village poultry [44]. This epidemiology of HPAI has changed in recent years with the emergence of the H5N1 clade 2.3.4.4b Eurasian lineage viruses. These have spread along recognised migratory bird routes across broad geographic areas of Asia, Europe, Africa, North America, South America, and recently, the Antarctica Polar Front. This has resulted in severe infection with disease and death in domestic and wild birds and mammals including major die-offs in seabird colonies and among endangered species [45,46,47].

Although avian influenza has a low prevalence, its outbreaks can have devastating effects. In addition to the direct impact on affected flocks in terms of illness and death, outbreaks also lead to consumer panic due to the fear of zoonotic transmission. This panic can cause a collapse in consumer demand for chicken products, further impacting smallholder chicken producers’ livelihoods [48]. Outbreaks also affect market access and may require costly control measures to be implemented.

Humans become infected from birds through contact with viruses in the secretions and excretions of infected birds, and while still a rare event, it has been more commonly reported in backyard or smallholder systems. The main symptoms are fever, cough, sore throat, runny or stuffy nose, muscle or body aches, headaches, fatigue, and shortness of breath or difficulty breathing. Since the first human outbreak of avian influenza H5N1 in Southeast Asia in 2003, 880 human cases of avian influenza have been reported globally up to the end of November 2023, with a case fatality rate of 52% [49]. The most serious public health concern with avian influenza is that the virus has the potential to evolve to be transmitted from human to human, which could precipitate a global pandemic, as occurred on three occasions with influenza during the 20th century [50].

Non-typhoidal salmonellosis, campylobacteriosis, and toxigenic colibacillosis are major contributors to the global human foodborne disease burden, including in LMICs [51,52], and as such are priority poultry zoonoses. The major reservoir of these pathogens are food animals, and most of this foodborne disease burden is of zoonotic origin. Studies in Africa showed poultry to be the major food animal reservoir of *Salmonella* and *Campylobacter* followed by pigs [42].

Zoonotic salmonellosis is mainly caused by *Salmonella enterica* subspecies Enterica serovars typhimurium and enteritidis. These serovars are commonly found in the gastro-intestinal tract (GIT) of chickens and usually cause no clinical disease in chickens. The contamination of chicken meat and eggs by GIT content from salmonella-infected chickens is the main source of infection in humans. In some instances, eggs may become infected transovarially from infected hens when the infection in the hen involves the ovaries. Humans acquire infections mostly via the ingestion of contaminated meat or eggs. The clinical signs in humans are fever, diarrhoea, and abdominal pain and are usually self-limiting, disappearing in two to seven days without treatment. But it may cause severe disease in individuals with weakened immunity. The prevention of chicken-derived salmonellosis in humans requires good hygiene practices on poultry farms and during live bird transport (to reduce shedding and cross-contamination between birds), the hygienic handling of eggs and meat during processing to prevent cross-contamination and environmental contamination, the discarding of cracked eggs, proper storage (refrigeration), and the adequate cooking of chicken meat and eggs at an appropriate internal temperature. The vaccination of poultry is also an important adjunct for controlling *Salmonella*, though not widely practiced in Africa. In some cultures, in Africa, the consumption of raw egg is a traditional treatment for illness (as was found in a study in Kampala [53]), and this risky practice could expose vulnerable people to Salmonellae and other hazards. Similarly, a survey in KwaZulu-Natal found raw eggs were consumed by parturient women to facilitate labour [54].

*Campylobacter jejuni* and *C. coli* are the major species associated with enteritis in domesticated animals and humans. Chickens are natural reservoirs of human-pathogenic species such as *C. jejuni*, *C. coli*, and *C. lari* but do not typically contract the disease themselves. The most dominant zoonotic species of chicken is *C. jejuni*. The common clinical symptoms in humans include diarrhoea (usually bloody), abdominal pain, fever, headache, nausea, and/or vomiting. Globally, *Campylobacter* spp. is the most common cause of bacterial gastroenteritis, and poultry are the main reservoir [55]. Less is known of the epidemiology in Africa, but a recent systematic review found *Campylobacter* spp. was common in Africa, with the highest prevalence being in West Africa [42]. Infection prevention measures in poultry flocks include biosecurity, such as cleaning and avoiding overcrowding. The measures of prevention for the transmission of zoonotic *Campylobacter* spp. to humans is similar to the salmonellosis prevention measures described above, such as the proper handling of eggs and meat during processing, storage, and cooking. However, controlling and preventing *Campylobacter* spp. contamination of poultry and poultry products are much more challenging.

As well as these poultry zoonoses of major public health concern, poultry can transmit zoonoses of less concern or whose importance is not known. Newcastle disease virus has also been reported to infect mammals including humans. Humans contract infections under high-viral-load exposure, but transmission from birds to humans is rare [56]. When the disease occurs in humans, it generally causes conjunctivitis and mild influenza-like symptoms and is self-limiting. It is generally not an important zoonosis. However, a lethal form of infection has been described in immunocompromised humans [56].

Avian chlamydiosis is a systemic bacterial infection caused by *Chlamydia psittaci*. It is primarily an infection of psittacine birds such as parrots but also causes disease in poultry. Among poultry, turkeys and ducks are more susceptible than chickens. The disease can be transmitted to humans, causing a zoonosis referred to as psittacosis or ornithosis. Apart from parrots, humans usually acquire infection from turkeys or ducks, but chicken-associated outbreaks of psittacosis have been reported [57], including one in a chicken slaughterhouse [58]. Symptoms in humans can vary from mild to severe, including fever, headache, muscle ache, cough, colitis, and diarrhoea, and usually present as community-acquired pneumonia. Psittacosis is a rare but possibly under-diagnosed disease responsible for around 1% of CAP cases; few of these cases are acquired from chickens.

Fowl mite (*Dermanyssus gallinae)* is a nonburrowing, bloodsucking mite that parasitises poultry and other bird species. It can also infest humans. The mite migrates from bird nests into buildings and attacks humans. In humans, it causes painful bites; pruritus (most intense at night); allergic dermatitis; rash; papules; vesicles; and lesions, often on the backs of hands and forearms of poultry workers. To prevent mite attack, people should wash their hands and wear gloves and protective clothing when handling poultry and material in poultry environments; environmental hygiene is also important. Fowl mite is a common but not serious cause of human ectoparasitosis.

Poultry erysipelas is a bacterial disease caused by *Erysipelothrix rhusiopathiae.* It is a sporadic cause of acute septicaemia and, less rarely, chronic skin and joint lesions in chickens. Humans acquire erysipeloid from direct contact or ingestion. This usually presents as acute, localised cellulitis and is a rare disease. Avian tuberculosis occasionally causes cases in elderly or immunocompromised people. It is not well documented in Africa. *Lophophyton gallinae* (*Microsporum gallinae*) is a zoophilic fungus that causes ringworm in chickens and related species, and occasionally in humans. It is not a serious zoonosis.

Avian colibacillosis is an infectious disease of birds caused by *Escherichia coli*, which, along with avian salmonellosis, is considered one of the principal causes of morbidity and mortality in poultry. Toxigenic *E. coli* is a major source of human disease, and Shiga toxin-producing *Escherichia coli* (STEC) (also known as verocytotoxin-producing *E. coli* (VTEC)) is zoonotic. Ruminants are considered the most important animal reservoirs, but STEC has been detected in broilers. However, STEC is rare in Africa. In addition, avian colibacillosis might be a zoonotic risk due to the high genetic similarity between certain isolates and *E. coli* causing urinary tract infections (UTIs) in humans, and is also a major concern for the transmission of antimicrobial resistance (AMR) genes [59].

## 7. The Prevention of Disease in Poultry: Biosecurity, Good Agricultural Practices, Vaccination, and the Use of Disease-Resistant Breeds

“An ounce of prevention is worth a pound of cure”, and the major focus in disease management should be on the prevention of disease occurrence and transmission rather than treating after the disease has occurred. This is especially true for poultry as the value of an individual bird is small, yet the costs of accurate diagnosis and correct treatment are substantial. To reduce risk to human health, it is better to control diseases in the animal host rather than attempting to reduce transmission to people or treat in the human victim [60]; hence, this section focuses on the control of diseases with public health implication in poultry.

Poultry diseases have been historically prevented in the intensive poultry industry through the use of prophylaxis (administering antimicrobials before exposure to disease) or metaphylaxis (administering antimicrobials after disease detection to reduce spread). However, there is a growing concern about these practices contributing to drug resistance in humans, livestock, and plants. To address this issue, alternative approaches to using antimicrobials for disease prevention are being explored. Some of the alternatives to the prophylactic use of antimicrobials include prebiotics, probiotics, bacteriophages, phytochemicals, enzymes, and antimicrobial peptides [61,62]. These non-antimicrobial options show promise in controlling poultry diseases without the associated risks of promoting antimicrobial resistance; however, many of these alternatives are complex in application and still at an early stage of use in high-income countries. They are unlikely to be adopted in small-scale poultry systems in Africa in the near future.

Meanwhile, preventing infectious diseases in smallholder poultry systems in Africa can be achieved through various measures that fall into broad categories. Most important are biosecurity, husbandry or management practices, vaccination, and the selection of resistant birds.

### 7.1. Biosecurity

The World Organization for Animal Health (WOAH, formerly OIE) Terrestrial Animal Health Code (2022) defines biosecurity as “a set of management and physical measures designed to reduce the risk of introduction, establishment and spread of animal diseases, infections or infestations to, from and within an animal population” [63]. Biosecurity is the main strategy for disease control in intensive poultry production, but much less effort has been made to develop biosecurity measures appropriate for use in resource-limited settings. A systematic literature review found only 23 documents giving precise biosecurity-related recommendations at flock level for smallholder flocks and no general guidelines for backyard poultry-related biosecurity in LMICs [16,64,65]. Moreover, few documents were found about the impact of measures in backyard settings, and none gave any evidence of their feasibility and effectiveness.

The biosecurity measures that have been recommended specifically for backyard or smallholder farmers fall into three broad categories: (i) isolation; (ii) traffic control; and (iii) sanitation [64] (Table 3).

Isolation is a key biosecurity measure for preventing disease transmission among poultry flocks as infectious disease is typically spread from sick or dead birds or their excretions and secretions. Isolation involves keeping infected animals separated from the rest of the flock until the pathogen is eradicated. Additionally, to maintain a closed flock and avoid introducing new birds, precautions must be taken. Even if new birds appear healthy, they can still carry diseases that may affect the existing flock. It is especially risky to introduce birds from unknown sources. If new birds must be added, they should come from trusted suppliers and undergo quarantine before joining the flock. Another approach to disease control is the “all-in, all-out” or single-batch system, where all birds are acquired at the same age and from the same source, without adding new birds. This management strategy allows for proper sanitation and a resting period between batches. However, it may not be feasible for farmers who need to harvest birds based on demand or rely on hens breeding to replenish the flock.

Traffic control is a crucial biosecurity measure implemented to minimise the transmission of infections through the movement of animals, people, vehicles, and materials within or into a flock. To maintain strict control, only designated workers should have access to the flock premises, whether they are housed or fenced. In the event of visitors, they should be permitted entry only after thorough hand cleaning and the changing or disinfection of footwear and clothing. Caution must be exercised for visitors who have been in contact with other birds, such as poultry farms, pet birds, or wild birds. Workers engaged in highly contaminated tasks like manure cleaning or handling sick and dead birds should undergo proper cleaning and disinfection before transitioning to clean activities such as feeding, watering, and egg handling.

Sanitation involves the cleaning and disinfection of people, materials, equipment, vehicles, and premises. It is crucial for individuals responsible for poultry care to adhere to strict hygiene practices, including regular handwashing before and after attending to the birds. Materials and equipment brought onto the farm should undergo thorough cleaning and disinfection. Equipment used for tasks involving sick or deceased birds must be cleaned and disinfected extensively before they are used for other operations within the flock. When it comes to vehicles, they should also be cleaned and disinfected before entering or approaching the poultry premises. A high-pressure spray should be utilised for cleaning, followed by thorough disinfection. It is advisable to situate vehicle parking areas as far away from the premises as possible. The prompt removal and immediate disposal (burial or burning) of dead birds are essential. The designated dead bird disposal pit should be located at a considerable distance from the poultry premises.

While biosecurity is generally agreed as essential for preventing and controlling infectious diseases in poultry flocks, as mentioned earlier, implementing biosecurity can be challenging for smallholder poultry producers in Africa and elsewhere who have limited resources in terms of money but also time and labour. Isolation measures like housing birds are often not practical [16,66]. Moreover, applying inappropriate biosecurity measures in smallholder poultry may generate little benefit for household income and fail to be cost-effective [23]. This implies that more resource-intensive biosecurity may require the subsidisation of biosecurity costs when justified by externalities such as the public health benefits of a reduction in transboundary spread, or where the development of smallholder poultry has livelihood and economic benefits that justify the costs of subsidising biosecurity.

Some biosecurity measures are relatively more adaptable to the extensive production system than others. Biosecurity measures designed for smallholder extensive production are outlined in a manual for improving village chicken production [65].

### 7.2. Good Husbandry or Good Agricultural Practices

There is some overlap between biosecurity and good husbandry or Good Agricultural Practices (GAPs). While biosecurity primarily aims to prevent the entry, establishment, and spread of diseases, GAPs focus on sustainable and efficient agricultural practices that optimise productivity while minimizing environmental impacts, optimising animal welfare, and ensuring the safety and quality of the products. GAPs can also encompass health and occupational safety by ensuring production practices do not put the health of workers at risk.

As such, GAPs also play a crucial role in disease prevention in both poultry and people. They include implementing appropriate flock density, providing clean and appropriate housing, ensuring access to clean water and a balanced diet, practicing humane handling, and managing waste and manure properly to minimise disease transmission. An example from Cambodia, relevant to Africa, involved a training and knowledge-sharing program on good poultry production and health practices and the development of poultry breeding and poultry-fattening units in village communities. This reduced the mortality rates of indigenous chickens from 80 to 5–10 percent and improved producers’ income by 150–200 USD/month from poultry production [67].

Integrated pest management is a type of GAP of especial relevance to tropical producers. In poultry husbandry, IPM has been used, for example, in the control of red mites, the most damaging ectoparasite of laying hens [68]. This consists of eight steps, in which the prevention of introduction of mites in poultry houses and the monitoring of the pest are essential for sustainable control. Primarily, strategies and tactics that are safe for the environment and do not include chemicals are employed to prevent and control the pest species. Chemical treatments are only used as a last resort once non-chemical ones have failed and an action threshold has been reached.

However, smallholder farmers face similar challenges in implementing GAPs to those they face in biosecurity. Farmers usually lack access to information or advice on GAPs [69] and, in addition, may lack the money, time, and human capital needed to implement GAPs.

### 7.3. Vaccination

Vaccination plays a crucial role in preventing vaccine-preventable infectious diseases in large-scale poultry production. By exposing the host to a component of the disease-causing agent, vaccines stimulate the immune system and provide targeted prevention for specific diseases. In theory, while biosecurity measures designed for intensive poultry production may not be easily applicable to extensive smallholder poultry systems, vaccination could be a more feasible, effective, and affordable strategy in this setting. In practice, the uptake of vaccination by smallholders in Africa has been low. For example, the estimated vaccination coverage for poultry (mainly against Newcastle disease) is about 36% in Ethiopia [13], about 35% in Uganda [70], and 26% in Tanzania [71]. While this coverage will help to prevent disease at the level of participating households, it is too low for herd immunity, so diseases persist even in the presence of vaccination.

In the context of projects, the introduction of vaccination, especially for Newcastle disease, has had significant positive impacts on reducing chicken mortality and providing economic benefits for smallholder chicken farmers [72,73]. Without outside intervention, or after the end of the project, the benefits are less clear. For example, one study investigated villages which had benefited from a well-organised vaccination program in Tanzania some years after it had been handed over to the government to manage [74]. Only 42% of households in the program villages used the vaccine, although this was more than twice as many as in the non-program villages, and while losses from Newcastle disease (as reported by farmers) were lower in the program villages (an average of 14 birds per year versus 20 birds per year), they were still considerable.

There are many limitations to vaccination in smallholder settings. The requirement to maintain vaccines in a cold chain, ensuring they are kept cold from production to delivery, can be challenging in areas with limited access to reliable electricity, transportation, and refrigeration equipment. Additionally, the small and multi-age flock structure of extensive smallholder poultry systems presents challenges for cost-effective vaccine delivery. Unlike intensive systems with larger numbers of birds being vaccinated at once, smallholder systems deal with smaller flock sizes and variable vaccination needs, making economies of scale less attainable [75]. Most vaccines are sold in vials of several hundred doses, which must be used in a short time after reconstitution, an additional challenge for smallholder poultry farmers. There have been some efforts to develop vaccines in affordable package sizes and longer post-reconstitution viability [76], but these are not widely available in LMICs. In practice, vaccinators try to cover many farms in a short time, which probably leads to some vaccine failures [74].

The single most common disease for which vaccination is widely practiced in the smallholder setting is Newcastle disease. There are different effective vaccines available for Newcastle disease including the thermotolerant vaccine I-2, which can be used in the absence of well-maintained cold chains [73]. Some other vaccines for important poultry diseases like avian influenza, though available, are not well suited for smallholders. The currently available vaccines for avian influenza are expensive, need frequent updating of viral subtypes included in the vaccines to ensure they match circulating subtypes, require frequent vaccination, and are injectable [77]. All these attributes are not favourable for smallholder application. Some chicken vaccines are given into the egg (i.e., in ovo), others in drinking water, or as sprays which are logistically easier when performed at scale compared to injection. Even in commercial flocks in Nigeria, influenza virus antigens have been detected in flocks with vaccination programs, suggesting vaccination failure, mismatch with circulating strains, or non-sterilising vaccination [78].

Poultry vaccination also has public health impacts by reducing the transmission of zoonotic pathogens and reducing the load of non-zoonotic pathogens with potential to mutate to zoonotic forms.

Technology developed during the COVID-19 pandemic has facilitated mRNA avian influenza vaccines that can be potent, cost-effective, more thermostable, and safe. Advantages include faster and cell-free manufacturing; the host cell produces the antigen after vaccination, with no risk of transportation into the nucleus and genomic integration, with the potential to develop “universal vaccines” effective against all strains. mRNA vaccines against avian influenza are currently undergoing research [79].

### 7.4. The Selection of Disease-Resistant Breeds

In addition to biosecurity and vaccination programs, the use of disease-resistant breeds offers another potential strategy to mitigate the impact of infectious diseases in smallholder chicken farming in Africa. Disease resistance refers to the ability of the host to resist infection or to suffer minimal adverse effects following infection, which is also known as disease tolerance. Africa has a wide genetic diversity of local chicken breeds and ecotypes, providing strong potential for the selection of disease-resistant traits [80]. Genomic studies have identified potential candidate genes and mutants associated with resistance to various viral and parasitic diseases, suggesting the possibility of using genomic selection for disease resistance in African local chicken ecotypes [81].

Historically, the introduction of exotic genetics into smallholder, intensive chicken production has not been competitive in sub-Saharan Africa due to management issues and high feed, veterinary, and energy costs [82]. More recently, there has been some success in the introduction of tropically adapted, “improved” breeds (TAIBs) that are dual-purpose (i.e., simultaneously supply meat and eggs), which require modest management conditions and are less susceptible to common diseases [83]. However, the overall feasibility of adopting these breeds and their impacts on smallholders’ livelihoods remain unknown.

## 8. The Treatment of Poultry Diseases

The treatment of diseased birds is often not advised, except for parasitic infections. This is because there are few cost-effective treatments for viral diseases, and the use of antibiotics in treating bacterial diseases can foster the emergence of antimicrobial resistance in human and animal pathogens, considered one of the most serious public health problems of the age. In practice, antimicrobials are not well controlled in Africa, and while farmers have a low propensity to use vaccination, they have a high propensity to seek treatment for sick animals, and most smallholder farmers attempt to treat sick birds either with traditional or modern medicines or both [44]. Where modern medicines are used, this is usually without veterinary guidance. Indeed, many studies across Africa find antimicrobials are almost always purchased without prescription or veterinary diagnosis from informal or quasi-legal sources such as agricultural input shops, open markets, or community members [84,85,86,87,88]. Most studies find that farmers and many animal health service providers lack awareness about correct dosage, antimicrobial residues, withdrawal periods, and antimicrobial resistance; widespread therapeutic use of antimicrobials in smallholder poultry is likely to contribute to antimicrobial resistance in people and livestock [89,90,91]. However, treatment by lay people is often perceived as sufficiently successful as to motivate persistence of this practice, especially given the low availability and high cost of veterinary care [92]. And, while most farmers with sick birds seek treatment, smallholder farmers in Africa tend to rely more on traditional medicine and have less use of antibiotics for prophylactic, metaphylactic, and growth promotion, and so use fewer antibiotics per bird than commercial farmers. For example, while traditional farmers in Sudan relied heavily on antimicrobials to control disease, there was limited prophylactic use and no reported use for growth promotion among smallholders [90].

In smallholder flocks, antimicrobials are typically administered in water or feed [86]. This can be a challenge for birds which are kept in scavenging systems. In this system, chickens could be restricted from scavenging for some hours before treatment to make them hungry and thirsty, and then the medicine is given through water or feed as recommended by the manufacturer. Generally, mass treatment through water and feed needs careful attention to ensure the proper dose is delivered to each bird.

While antimicrobial treatment is often not advised for smallholder poultry, treatment for endoparasites and ectoparasites is recommended. For coccidiosis and helminths, anticoccidial and anthelminthics can be given with feed or water. Pesticide (insecticide for lice and fleas and acaracide for mites) spray can be applied both on the birds and shelter. Some medicine such as ivermectin could be applied as pour on the birds both for ectoparasite and endoparasites. Care is needed in the application of pesticides as they can easily poison birds. A general piece of advice for any treatment in chickens is to strictly adhere to the recommended application method and dose of the medication.

Ethnoveterinary medicine (EVM) or traditional medicine is locally derived knowledge, skill, methods, practices, and beliefs of people in caring for, healing, and managing animals. These indigenous practices have been applied for centuries and have been passed down orally from generation to generation. Ethnoveterinary practices (EVPs) include the treatment and prevention of diseases, drug preparation, ectoparasite and endoparasite control, fertility or production enhancement, and bone-setting activities. EVPs are accessible and are mostly easy to prepare, cheap, and environmentally friendly. They can be easily adopted since they are part of the culture of the people. These practices can be used as alternatives or be complementary to modern health care approaches, especially in remote or less accessible areas. However, there are still concerns about the efficacy, safety, dosage, hygienic status, and standards of these practices. However, many see EVPs as a potential source of solutions to animal health problems if thorough evaluation, standardization, or optimization is performed [93,94].

It is beyond the scope of this short description of ethnoveterinary medicine to deal with individual products used as, firstly, they are numerous, and secondly, in most African countries, unlike South Asia, no ethnoveterinary products have been formally registered. Those requiring more information may consult the recent reviews showing that ethnoveterinary practices may play a role in improving the health and productivity of poultry in Africa [93,94,95,96,97].

## 9. Conclusions

Health losses pose significant challenges in smallholder poultry production systems worldwide, including Africa, where a wide range of highly prevalent diseases exist. Extensive smallholder systems are particularly prone to predation and other external forces. Infectious diseases in poultry not only affect animal health but also have implications for public health, including zoonotic transmission, antimicrobial resistance, and undernutrition.

Unfortunately, smallholder poultry producers face additional obstacles due to the lack of organised veterinary and animal health services specifically catering to their needs. The historical prioritization of animals of high economic value and the ability to pay for veterinary services has led to neglect in poultry health, with a greater emphasis on the health of cattle and small ruminants. This prioritization disproportionately affects women, who often own smallholder and indigenous poultry, while ruminants are more commonly owned by men.

Implementing appropriate biosecurity measures and Good Agricultural Practices can significantly reduce health losses in semi-intensive poultry systems. However, these measures pose challenges and can be costly for traditional smallholder chicken husbandry. As a result, vaccination becomes a widely suitable and effective approach to combat major infectious diseases. Nevertheless, vaccination presents its own difficulties due to the small size and varied structure of flocks, the short lifespan of individual poultry, and the lack of infrastructure in rural areas with limited resources. Additionally, while smallholder farmers demonstrate a strong inclination to seek treatment for sick poultry, limited availability, past vaccination failures (due to poor handling, lack of cold chains, poor technique, and vaccination in the face of outbreaks), and high time discounting may reduce their willingness to invest in more economically efficient vaccination practices. There is optimism from some that the selection of disease-resistant poultry breeds could play a significant role in reducing the impact of some diseases on smallholder chicken production in Africa (although there are currently no breeds resistant to HPAI or Newcastle disease). However, the adoption of these disease-resistant breeds is still in its early stages and requires further evaluation and appropriate support.

In conclusion, disease remains an enormous source of loss in smallholder poultry systems in Africa, jeopardises commercial production in Africa and elsewhere, and has a wide range of direct and indirect negative impacts on human health and well-being, as well as livelihoods. Many potential solutions exist, and there have been successes. But they are typically small scale, and only operate when projects and programs are in place. Most African countries lack sustainable and scalable solutions needed to support and develop the smallholder poultry systems which dominate the continent’s poultry sector.

## Figures and Tables

**Figure 1 foods-13-00411-f001:**
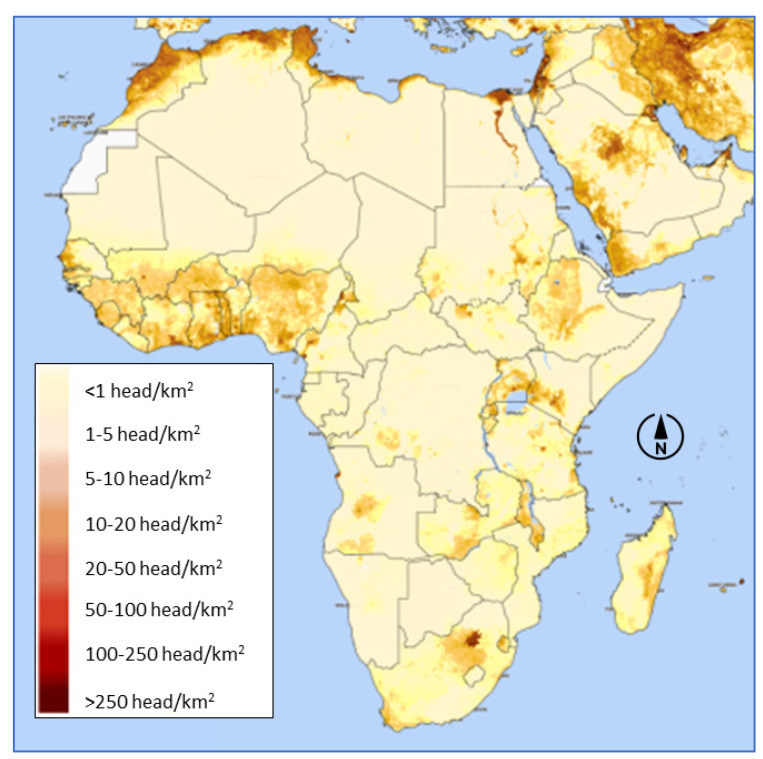
Chicken density in Africa—Map produced by FAO Hand-in-Hand Geospatial Platform Fri, 15 Dec 2023 [6].

**Table 1 foods-13-00411-t001:** Relative importance of major infectious diseases in different chicken genetic types and production systems.

Diseases	Genetics (Indigenous vs. Exotic)	Production System (Scavenging vs. Small-Scale Semi-Intensive)
Newcastle disease *	Both indigenous and exotic breeds are susceptible and can suffer high mortality.	The disease is common in unvaccinated birds, including scavenging birds, which too often have limited vaccination.
Highly pathogenic avian influenza *	High mortality is observed in all chicken breeds.	Important in intensive systems which keep large numbers of chickens.
Infectious bursal disease	Can affect both local and exotic breeds, but high mortality is often seen in exotic breeds.	Causes high mortality in intensively kept young chickens. Less common in multi-aged chickens kept under traditional scavenging systems.
Marek’s disease	Can affect both indigenous and exotic breeds.	Causes high mortality in intensive systems. Less common in multi-aged chickens kept under traditional scavenging systems.
Fowl cholera	Mild in indigenous breeds and often subclinical but causes mortality in exotic chickens.	More important in intensively managed chicken systems, which often keep exotic breeds.
Salmonellosis *	Mild in indigenous breeds.	Important in intensively managed chicken systems which keep exotic breeds.
Coccidiosis	Can affect, and is prevalent, in both local and exotic breeds. The disease is often subclinical in indigenous breeds.	Important in intensive systems with deep litter housing. It can also be important in scavenging birds, depending on the season.
Helminth parasites	Can affect both local and indigenous breeds.	Higher helminth exposure in extensive scavenging systems.

* zoonotic or potentially zoonotic.

**Table 2 foods-13-00411-t002:** Major poultry zoonotic diseases in sub-Saharan Africa.

Diseases	Presence and Prevalence	Production System (Scavenging vs. Small-Scale Semi-Intensive)
Avian influenza	Non-significant regional differences in prevalence were reported. Most outbreaks in West Africa [35].	Smallholder flocks are considered high risk because of low biosecurity and higher contact with wild birds. Village chicken flocks in remote areas away from wetlands are probably not high-risk.
Campylobacteriosis	Reported prevalence highest in Central Africa (91%), followed by Eastern, Southern and Western [42].	Commercial flocks at higher risk because of higher stocking density.
Salmonellosis	Reported prevalence highest in Southern Africa (28%), followed by Central, Eastern, and Western [42].	Larger farms have increased occurrence, persistence, and spread of *Salmonella.* Layer at higher risk than broiler.

**Table 3 foods-13-00411-t003:** Biosecurity for smallholder poultry.

Practices	Components	Intensive Production Systems/Specialised Backyard Systems	Extensive Production Systems
Isolation	Set-up	Keep housing away from public roads and stagnant water sources. Maintain a perimeter barrier. Use solid roofs and sides to prevent contact with infected wild birds or their droppings. Provide food and water only in covered areas; cover stored food. Ensure birds have adequate space, light, and ventilation.	Provide separate night housing for different poultry species. Chickens should not be bought from markets or neighbouring villages at times of the year when outbreaks of disease such as Newcastle Disease are common.
	Birds	Keep different ages and species separate. Avoid introducing new birds to existing flocks. Buy from reputable sellers. Follow approved vaccination schedules. Quarantine new birds. Quarantine sick birds.	Encourage separation between animal species and between animals and humans; waterfowl should be separated from chickens and turkeys. Avoid introducing new birds of unknown origin or from a sick flock into the “home” flock. Keep new or sick birds separate from the flock for 2 weeks. Vaccinate against key endemic vaccine-preventable diseases. Provide supplementary feeding when necessary to promote good health and a strong immune system.
	Pests	Practice rodent and insect control. Line gravel or sand outside houses and keep grass short.	Store chicken feed safely away from rodents and wild birds.
Traffic control	Daily routine	Wash hands before and after handling birds. Use rubber boots and protective clothing. Have disinfectant footbaths at the entrance to pens.	Wash hands with soap after handling birds, especially from other flocks.
	Visitors	Try to keep people away from your birds. Provide clean clothes and foot protection for visitors. Cover roads with sand or gravel. Clean vehicle tires before and after visits.	Avoid the “home” flock coming into contact with visitors, cages, or animals from an area where there is a disease outbreak in poultry.
	Business	Avoid sharing equipment with other backyard owners. Conduct business by mobile phone where possible.	If buyers come to the farm, keep them away from unsold birds.
Sanitation	Waste management	Wear gloves when handling waste. Remove and dispose of manure before adding new birds. Use composting to dispose of manure.	Clean chicken houses, troughs, and nests regularly. Regularly clean out and dispose of manure, and preferably stack for at least 3 weeks.
	Carcass disposal	Dead poultry may be disposed of by burying, composting, or incineration. Dead rodents and wild birds should be buried away from your flock.	Dispose of sick and dead animals and infected materials correctly, and clean and disinfect/decontaminate thoroughly. In villages where birds are dying of disease, no birds should be slaughtered for consumption.
	Decontamination	First clean to physically remove dirt which can block the disinfectant from the germs. Then, perform chemical disinfection to destroy pathogens.	Always scrub cages, egg trays, etc., with disinfectant or detergent and allow to dry before bringing them onto the farm. Manure, dirt, feathers, etc., stop the disinfectant working properly. If disinfectant is not available, items can be placed in a sealed black plastic bag in direct sunlight for 1 day so that the high temperature inside can inactivate disease agents. Slaughter only healthy birds from healthy flocks for consumption—immerse the bird in boiling water for a minute before plucking the feathers to inactivate any infectious agents on the outside of the bird.

Source: authors and Ahlers et al., 2009 [65].

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
