# Peer review of "The Public Health Importance and Management of Infectious Poultry Diseases in Smallholder Systems in Africa"

_foods, 2024, doi:10.3390/foods13030411_

Round 1

Reviewer 1 Report

Comments and Suggestions for Authors

The review manuscript titled "Prevalence, Impact, Public Health Importance, and Management of Infectious Poultry Diseases in Smallholder Systems in Africa" is well-written and contains valuable insights and viewpoints. However, there are several concerns that need to be addressed:

Ø  Address the procurement of vaccines by smallholder farmers in Africa. How do they acquire vaccines, considering they usually come in large doses? Are they procuring vaccines in bulk and sharing them? Discuss the necessity of small-dose vaccines (e.g., 10 doses or lower) in the context of smallholder poultry farming.

Ø  The manuscript repeatedly mentions the risks of antimicrobial resistance due to treatment. Please elaborate on the use of antimicrobials as prophylactic measures or growth promoters in African poultry farming practices.

Ø  Revise the phrasing in line 22 regarding antimicrobial resistance risks for clarity and precision.

Ø  Clarify the meaning in line 74: "disease or threats of disease by using antimicrobials?"

Ø  Eliminate redundancy: Lines 89-90 contain a definition of disease that is already stated in lines 79-80. Remove the repetition for conciseness.

Ø  Clarify the statement in line 92: "Takes place often?" This phrase lacks context; provide additional information or rephrase it for clarity.

Ø  Consider adding a table detailing previously recorded disease outbreaks in African poultry flocks. Additionally, a separate table may be included for zoonotic events related to poultry diseases.

Ø  Address the lack of data on ethnoveterinary treatment options or Indigenous Traditional Knowledge (ITK) practices followed by poultry farmers in Africa. If the authors do not have sufficient data on this topic, exclude from the manuscript.

Ø  Number the subsections within the manuscript to improve organization and readability.

Comments on the Quality of English Language

Minor editing of English language required

Author Response

The review manuscript titled "Prevalence, Impact, Public Health Importance, and Management of Infectious Poultry Diseases in Smallholder Systems in Africa" is well-written and contains valuable insights and viewpoints. However, there are several concerns that need to be addressed:

Thank you for your review. Please find our response in italics.

Ø  Address the procurement of vaccines by smallholder farmers in Africa. How do they acquire vaccines, considering they usually come in large doses? Are they procuring vaccines in bulk and sharing them? Discuss the necessity of small-dose vaccines (e.g., 10 doses or lower) in the context of smallholder poultry farming.

This is an important point. We have added “Most vaccines are sold in vials of several hundred doses which must be used in a short time after reconstitution, an additional challenge for smallholder poultry farmers. There have been some efforts to develop vaccines in affordable package sizes and longer post-reconstitution viability (Lal et al., 2014), but these are not widely available in LMICs. In practice, vaccinators try to cover many farms in a short time, which probably leads to some vaccine failures [56].”

Ø  The manuscript repeatedly mentions the risks of antimicrobial resistance due to treatment. Please elaborate on the use of antimicrobials as prophylactic measures or growth promoters in African poultry farming practices.

This was reported in several of the studies cited and we have expanded as an example “For example, while traditional farmers in Sudan relied heavily on antimicrobials to control disease, there was limited prophylactic use and no reported use for growth promotion [71].”

Ø  Revise the phrasing in line 22 regarding antimicrobial resistance risks for clarity and precision.

We have clarified by adding “and treatment for bacterial diseases risks antimicrobial resistance from imprudent antimicrobial use

Ø  Clarify the meaning in line 74: "disease or threats of disease by using antimicrobials?"

We have clarified as follows “farmers often respond to disease or concern that diseases are spreading locally by using antimicrobials”

Ø  Eliminate redundancy: Lines 89-90 contain a definition of disease that is already stated in lines 79-80. Remove the repetition for conciseness.

We have removed the redundancy.

Ø  Clarify the statement in line 92: "Takes place often?" This phrase lacks context; provide additional information or rephrase it for clarity.

We have rephrased for clarity “the environment where smallholder poultry production are farmed often lacks adequate sanitation”

Ø  Consider adding a table detailing previously recorded disease outbreaks in African poultry flocks. Additionally, a separate table may be included for zoonotic events related to poultry diseases.

This would be a useful addition but we could not find comprehensive literature recording disease outbreaks in backyard poultry in Africa. Table 1 provides a list of important diseases and we have added information on which are zoonotic.

Ø  Address the lack of data on ethnoveterinary treatment options or Indigenous Traditional Knowledge (ITK) practices followed by poultry farmers in Africa. If the authors do not have sufficient data on this topic, exclude from the manuscript.

We now mention that because of scope and word limit there is not a detailed consideration of ITC in African backyard poultry, but direct the reader to recent reviews for further information “Those requiring more information may consult the recent reviews showing ethnoveterinary practices may play a role in improving the health and productivity of poultry in Africa [76-78].”

Ø  Number the subsections within the manuscript to improve organization and readability.

We leave this to the journal typesetters.

Reviewer 2 Report

Comments and Suggestions for Authors

Dear Authors,

The review paper entitled “Prevalence, impact, public health importance and management of infectious poultry diseases in smallholder systems in Africa” is appropriately well written, developed and structured by Grace et al. in suitable English with a clear structure. They reviewed current knowledge and recent publications on poultry diseases in smallholder systems and the preventive strategies against the diseases. This manuscript is very interesting; however, there are some concerns regarding this paper. My main concern is that the scope of this study is so far from the Foods journal. This paper should be rewritten regarding the food microbiology aspect and point of views. Most parts of this paper have been structured and developed for animal sciences, agricultural sciences and veterinary medicine journals. Disease prevention and disease treatment sections should be removed; on the other hand, contamination prevention and antimicrobial processes should be added accordingly. Only zoonotic diseases can be discussed in this paper (Table 1). All parts of the paper should be rewritten related to food and food microbiology (also the introduction section). Otherwise, this manuscript cannot be considered for publishing in the Foods journal and I recommend other journals such as Poultry MDPI. Therefore, I recommend a major revision of this manuscript. If the authors cannot rewrite and restructure the paper, I recommend they withdraw the submission and submit this valuable paper to other journals such as Poultry or Veterinary Sciences in MDPI publishing. 

Author Response

Thank you for your review. The authors believe the review is in scope as did the other two reviewers.

We agree that some of the writing did not sufficiently emphasise the public health aspects which are the focus of the paper, and especially did not clarify that the topic was public health in its broadest sense, not just zoonotic disease.

We have revised to better reflect this. The title now reads “Public health importance and management of infectious poultry diseases in smallholder systems in Africa”. We have clarified our aim in the first paragraph “This comprehensive review aims to provide an in-depth understanding of the public health implications of important poultry disease in Africa, and how these diseases can be managed to reduce human health impact. The focus is on smallholder systems, including scavenging extensive systems and small-scale semi-intensified or intensified systems, which remain the predominant forms of poultry production in Africa, despite rapid growth of intensive systems. We consider broad public health implications that are not limited to zoonoses but include antimicrobial resistance and food and nutrition security”

We have expanded the section on poultry disease to emphasise this link between non-zoonotic disease and public health as follows “Productivity loss and culling in response to outbreaks reduces the availability of nutritious meat and eggs, which can have significant impacts in vulnerable communities. This was documented when an avian influenza outbreak resulted in mass culling of chickens in Lower Egypt but not Upper Egypt. Decreased dietary diversity, reduced poultry consumption, substitution of nutritious foods with sugary foods and increased stunting was seen in Lower Egypt but not Upper Egypt (Kavle et al., 2016).” And “Diseases also directly impact poultry welfare and, because farmers often respond to disease or concern that diseases are spreading locally by using antimicrobials, even non-zoonotic poultry diseases indirectly contribute to antimicrobial resistance in both people and animals”

Reviewer 3 Report

Comments and Suggestions for Authors

This is an interesting summary of poultry diseases in Africa although the title is a bit misleading. '........Prevalence, impact, public health importance and management of infectious poultry diseases in smallholder systems in Africa....' Table 1 provides an overview of a number of common poultry diseases but there is no analysis provided about which diseases are most prevalent in different regions in Africa. Most of the data given seems to relate to sub-saharan countries in Africa ? It would be good to provide some data to show the typical poultry management systems in different regions and within regions as this would provide some context for the management recommendations. The different cultural perspectives and food preferences/cooking & food handling preferences would also be interesting. Currently the 'review' doesn't add a lot to the current state of knowledge on the topic but it could be improved if the above points are taken into consideration. Has any comparative analysis been done ? It would also be good to provide some economic data for the poultry sector in different parts of Africa and an over view of the extent of small holder poultry farming.

Comments on the Quality of English Language

NA

Author Response

Thank you for your review. Our responses are in Italics.

his is an interesting summary of poultry diseases in Africa although the title is a bit misleading. '........Prevalence, impact, public health importance and management of infectious poultry diseases in smallholder systems in Africa....'

We have changed the title to be more informative “Public health importance and management of infectious poultry diseases in smallholder systems in Africa”

Table 1 provides an overview of a number of common poultry diseases but there is no analysis provided about which diseases are most prevalent in different regions in Africa. Most of the data given seems to relate to sub-saharan countries in Africa ?

We have clarified that because the focus is smallholder systems and these predominate in sub-Saharan Africa most of the data refers to sub-Saharan Africa.

It would be good to provide some data to show the typical poultry management systems in different regions and within regions as this would provide some context for the management recommendations.

Other reviewers suggested there was too much emphasis on poultry management and there should be more focus on public health. We have tried to balance these opposing recommendations but as the Special Edition is on Public Health we have chosen to emphasise public health rather than management aspects.

The different cultural perspectives and food preferences/cooking & food handling preferences would also be interesting.

We have expanded on the reference to a study in Kampala where raw eggs were used as medicine and added another study from KwaZulu-Natal which found raw eggs were believed to facilitate giving birth.

Currently the 'review' doesn't add a lot to the current state of knowledge on the topic but it could be improved if the above points are taken into consideration. Has any comparative analysis been done ?

The review does not aim to provide new data but rather to collate existing data on public health implications of poultry disease in smallholder systems in Africa. Smallholder systems are the most important production system in Africa and have major public health implications but we could not find any previous reviews on this topic so thought it would be of interest.

It would also be good to provide some economic data for the poultry sector in different parts of Africa and an over view of the extent of small holder poultry farming.

As mentioned, another reviewer suggested there was too much emphasis on poultry management and there should be more focus on public health. We have tried to balance these opposing recommendations but as the Special Edition is on Public Health we have chosen to emphasise public health rather than management aspects.

Round 2

Reviewer 2 Report

Comments and Suggestions for Authors

Dear authors, 

Thank you for your response. I have no more comments. 

Reviewer 3 Report

Comments and Suggestions for Authors

The clarified scope has improved the manuscript

Comments on the Quality of English Language

Minor edits